# Identification of *Fusarium oxysporum* f. sp. *lactucae* Race 1 as the Causal Agent of Lettuce Fusarium Wilt in Greece, Commercial Cultivars’ Susceptibility, and Temporal Expression of Defense-Related Genes

**DOI:** 10.3390/microorganisms11041082

**Published:** 2023-04-20

**Authors:** George T. Tziros, George S. Karaoglanidis

**Affiliations:** Laboratory of Plant Pathology, Faculty of Agriculture, Forestry and Natural Environment, Aristotle University of Thessaloniki, P.O. Box 269, 54124 Thessaloniki, Greece; gtziros@yahoo.gr

**Keywords:** *Lactuca sativa*, fusarium wilt, gene expression, defense responses

## Abstract

Fusarium wilt of lettuce is found throughout the world, causing significant yield losses. Lettuce is the most-cultivated leafy vegetable in Greece, affected by a large number of foliar and soil-borne pathogens. In this study, 84 isolates of *Fusarium oxysporum*, obtained from soil-grown lettuce plants exhibiting wilt symptoms, were characterized as belonging to race 1 of *F. oxysporum* f. sp. *lactucae* based on sequence analysis of the translation elongation factor 1-alpha (*TEF1-α*) gene and the rDNA intergenic spacer (rDNA-IGS) region. The isolates were also assigned to one single race through PCR assays with specific primers targeting race 1 and race 4 of the pathogen. In addition, four representative isolates were confirmed to be associated with race 1 based on the pathogenicity tests with a set of differential lettuce cultivars. Artificial inoculations on the most commonly cultivated lettuce cultivars in Greece revealed that the tested cultivars varied regarding their susceptibility to *F. oxysporum* f. sp. *lactucae* race 1. Cultivars (cvs.) “Cencibel” and “Lugano” were found to be highly susceptible, while cvs. “Sandalina” and “Starfighter” were the most resistant ones. Expression analysis of 10 defense-related genes (*PRB1*, *HPL1*, *LTC1*, *SOD*, *ERF1*, *PAL1*, *LOX*, *MPK*, *BG*, and *GST*) was carried out on artificially inoculated lettuce plants of the four above cultivars at different time points after inoculation. In resistant cultivars, a higher induction rate was observed for all the tested genes in comparison with the susceptible ones. Moreover, in resistant cultivars, all genes except *LTC1*, *MPK,* and *GST* showed their highest induction levels in their earliest stages of infection. The results of this study are expected to contribute to the implementation of an integrated management program to control Fusarium wilt of lettuce, based mainly on the use of resistant cultivars.

## 1. Introduction

*Fusarium oxysporum* f. sp. *lactucae* was first identified as the causal agent of Fusarium wilt of lettuce in Japan [1]. Thereafter, the pathogen has been found throughout the world [2], evolving into the most significant pathogen of lettuce cultivation [3]. The disease was first recorded in Europe in 2001 [4], causing significant yield losses of up to 70% in intensive cropping systems in Italy [5]. Nowadays, there are reports of this disease in seven more European countries: Belgium, England, France, Ireland, the Netherlands, Norway, and Spain [2]. *Fusarium oxysporum* f. sp. *lactucae* has been differentiated into four races to date: races 1, 2 and 3 were first identified in Japan [6,7], while isolates from Belgium [8], the Netherlands [9], England, Ireland [10], and Italy [11] have recently been assigned to the new race 4.

Using the vegetative compatibility group (VCG) approach, races 1, 2, 3, and 4 have been assigned to different somatic compatibility groups (VCG), named VCG 0300, VCG 0301, VCG 0302, and VCG 0303, respectively [12,13,14,15,16,17]. However, although these bioassays have been used extensively to define genetic relationships within the forma specialis *lactucae*, they have been replaced by less time-consuming molecular identification techniques.

Isolates within a forma specialis are related to their pathogenicity to a given host [18], and pathogenicity tests have been used to distinguish *F. oxysporum* formae speciales [16]. *F. oxysporum* f. sp. *lactucae* race determination is commonly based on the pathogenic ability of isolates using a set of differential lettuce cultivars [7,9,12]. Nonetheless, the results of the pathogenicity tests have to be supported by molecular analysis of the isolates, which enables race differentiation for a large number of isolates within a shorter time.

Molecular techniques have been developed to distinguish *F. oxysporum* f. sp. *lactucae* from other formae speciales in the *F. oxysporum* complex and to characterize the different races in forma specialis *lactucae* [19]. Sequence analyses of the mitochondrial small subunit (mtSSU), the translation elongation factor 1-alpha (*TEF1-α*) gene, and the nuclear ribosomal DNA intergenic spacer (rDNA-IGS) region have been used to determine diversity within *F. oxysporum* f. sp. *lactucae* [16]. Sequence analyses of the *TEF1-α* gene and IGS region have also recently been used to characterize the population structure of *F. oxysporum* f. sp. *lactucae* in California and Arizona [20], and in a study aiming to identify a new race (race 4) in Dutch isolates [9]. Moreover, additional specific molecular markers have enabled the identification of *F. oxysporum* f. sp. *lactucae* races. In particular, Pasquali et al. [5] and Gilardi et al. [9] have developed specific PCR primers, based on the inter-retrotransposon sequence-characterized amplified regions (IRAP-SCAR) technique, enabling the assignment of given isolates to race 1 and race 4, respectively.

*Fusarium oxysporum* is a ubiquitous root-infecting fungal pathogen that causes wilt disease in a wide range of plant species [21]. Hemibiotrophic pathogens, such as *F. oxysporum*, typically start their infection cycle with the biotrophic phase, establishing infection via the roots and travelling towards the vascular system of the plant [22,23,24]. Successful pathogens, such as *F. oxysporum*, have evolved mechanisms to overcome the relatively weak defense response that is induced by the plants upon perception of fungal elicitors in this stage of infection, and eventually colonize the plant [25]. Thereafter, the pathogen travels upwards through the vascular system, causing vascular wilting due to the accumulation of fungal mycelia and defense-related compounds in the xylem. As the infection progresses, *F*. *oxysporum* switches to a necrotrophic pathogen, causing foliar necrosis, lesion development, and eventual plant death [22,24]. This change in lifestyle implies that hemibiotrophic pathogens are able to take over the host signaling pathways [23]. The pathogen invasion most likely activates hormone-controlled defense pathways, such as systemic acquired resistance (SAR), which protects against subsequent infections [26]. SAR is mediated by the salicylic acid (SA) signaling pathway, although it is considered that jasmonate (JA) is also required in the initial stages [27].

Numerous anatomical and biochemical responses are involved in plants’ response to infection by *F. oxysporum*, e.g., modifying cell walls to block the pathogen, producing extracellular enzymes to directly degrade it, or activating the expression of specific genes through several signaling pathways and transcription factors [28]. More specifically, pathogenesis-related (PR) proteins, reactive oxygen species, and phytoalexins are expressed locally as well as systemically [29], while the ethylene (ET), ABA, salicylic acid (SA) and jasmonic acid (JA) signaling pathways have been reported to contribute to defense against *F. oxysporum* in the model plant *Arabidopsis thaliana* [30,31]. Moreover, it has been reported previously that the pathogen *Fusarium* sp. causes gene expression changes in a number of different defense response genes in several crops, including pigeonpea, chickpea, and castor [32,33,34]. However, to our knowledge, there are no data available on the expression changes in defense response genes during lettuce–*Fusarium oxysporum* f. sp. *lactucae* interaction. Nonetheless, a study focused on expression profiling using RNAseq and the sequenced lettuce genome identified a large number of differentially expressed genes involved in lettuce–*B. cinerea* interaction, associating them with the corresponding biological pathways in which they are involved [35]. More specifically, in the same study, the response of the selected genes was evaluated for a necrotroph (*B. cinerea*) and a biotroph (*Bremia lactucae*), highlighting that similar biological pathways are induced upon the inoculation of lettuce with necrotrophic and biotrophic pathogens. In addition, temporal and quantitative gene expression fluctuations were observed between one resistant and one susceptible lettuce cultivar following infection with *Sclerotinia sclerotiorum* [36].

Genes related to the induction of resistance in lettuce are the pathogenesis-related protein (*PRB1*) gene, associated with salicylic acid-mediated disease resistance [37], fatty acid hydroperoxide lyase (*HPL1*) and lipoxygenase (*LOX*), the two enzymes involved in the lipoxygenase (*LOX*) pathway to produce phyto-oxylipins acting as defense and signaling molecules [38,39], the sesquiterpene synthase (*LTC1*) gene, encoding a sesquiterpene synthase involved in the defense response against fungal pathogens [40], superoxide dismutase (*SOD*), a crucial enzyme protecting normal cells from reactive oxygen species (ROS) produced during many intracellular pathogen infections [41], ethylene response factor 1 (*ERF1*), which encodes a transcription factor that regulates the expression of pathogen response genes that prevent disease progression and whose expression can be activated by ethylene, jasmonate, or by both hormones synergistically [42], phenylalanine ammonia lyase (*PAL1*), the regulatory enzyme catalyzing the first step of the phenylpropanoid pathway [43], which biosynthesizes compounds with antimicrobial activity such as phytoalexins [44], the mitogen-activated protein kinase (*MAPK*) gene, involved in pathogen-induced defense signal transduction pathways [45,46], the β-glucanase (*BG*) gene, which has the ability to degrade fungal cell walls and may be involved in the defense mechanism of plants against pathogenic fungi [47], and a glutathione transferase (*GST*) gene that participates in the translocation of flavonoids, affecting disease resistance in plants [48].

Although Fusarium wilt is considered a common soil-borne disease of lettuce in Greece, no data are available regarding the forma specialis and, therefore, the races present in this crop. Hence, the objectives of this research were to (a) identify the isolates obtained from symptomatic plants using sequence analysis of the translation elongation factor 1-alpha (*TEF1-α*) and the nuclear ribosomal DNA intergenic spacer (rDNA-IGS) region, (b) assess the pathogenicity of selected isolates on a set of commercial and differential lettuce cultivars, (c) characterize some of the most commonly cultivated lettuce cultivars in Greece in terms of susceptibility/resistance to Fusarium wilt, and (d) compare the gene expression profiles of ten defense-related genes induced after the artificial inoculation of two susceptible and two resistant lettuce cultivars with the pathogen.

## 2. Materials and Methods

### 2.1. Pathogen Isolation

In June 2019, wilting symptoms were observed in two-month-old Iceberg lettuce plants in a commercial farm specialized in the cultivation of leafy vegetables (Vezyroglou Farm, Alexandria, Imathia, Central Macedonia, Greece). Affected plants showed stunted growth and leaf chlorosis, while the vascular tissues of taproots and crowns exhibited brownish discoloration. The disease was severe, resulting in the complete wilting and mortality of 50–60% of the soil-grown lettuce plants cultivated under nethouse conditions in an area of approximately 5 hectares.

For pathogen isolation, diseased plants were randomly selected at several sites of the surveyed area and transferred to the laboratory. The roots and crowns of symptomatic plants were surface-disinfested with a 1% sodium hypochlorite solution for 1 min and then washed three times with sterilized distilled water. Small pieces of necrotic vascular tissues were placed on potato dextrose agar (PDA; Oxoid, Thermo Fisher Scientific, Leiden, The Netherlands) amended with lactic acid, and incubated for 3 to 4 days at 23 °C in the dark. The yielded colonies were observed microscopically for morphological characteristics of *Fusarium oxysporum* [49]. In total, 84 single-spore cultures were obtained and cultures on PDA were deposited in the fungal collection of the Plant Pathology Lab, AUTh.

In addition to the isolates obtained from the field, a set of reference isolates were also used in our study: isolate Fus 1.39 (race 1) and Fus 1.01 (race 4) from Belgium, isolate 04750888 (race 4) from the Netherlands, isolates 231724 and 231725 (race 1) from Norway, and isolates MAFF 244120 (race 1), MAFF 244121 (race 2), and MAFF 2441222 (race 3) from Japan.

### 2.2. Molecular Identification of Fungal Strains

#### 2.2.1. DNA Extraction

Petri dishes (diameter 9 cm) containing ca. 20 mL PDA, overlain with a sterilized cellophane sheet (Gel drying frames, Sigma-Aldrich Chemie GmbH, Taufkirchen, Germany), were inoculated with mycelial plugs (4 mm in diameter) and incubated for 3–5 days in the dark at 23 °C. The mycelium was scraped from each plate, lyophilized, and ground to a fine powder. Genomic DNA was extracted from this material using a DNeasy Plant Mini Kit (Qiagen, Hilden, Germany) following the manufacturer’s instructions. The concentration of the extracted DNA was measured using a P330 nanophotometer (Implen GmbH, Munich, Germany).

#### 2.2.2. PCR Amplification

Initially, the translation elongation factor 1-alpha (*TEF1-α*) gene and the rDNA intergenic spacer (rDNA-IGS) region were amplified using the specific primers EF1/EF2 [50] and CNL12/CNS1 [16,51,52], respectively. Due to its large size, the IGS region was also amplified with the primer pair CNS12/RU46.67 [16]. The PCR reagents and conditions were as those described by Gilardi et al. [9]. Aliquots of the PCR products were loaded on 1.0% agarose gel in Tris-acetate-EDTA buffer with Midori Green Advance gel stain (Nippon, Düren, Germany). PCR products were purified with a PureLink PCR Purification Kit (Invitrogen, Carlsbad, CA, USA) and custom sequenced (CEMIA).

In addition, PCR was performed with the primers Hani3′/Hanilatt3rev [5] and FUPF/FUPR [9] to check whether the Greek isolates belonged to race 1 or race 4, respectively. DNA templates from the Greek isolates GTFus1-4 and the reference isolates, mentioned in Section 2.1, were used in the PCR assays. The PCR reagents and conditions for discrimination of race 1 and race 4 were as those described by Cabral et al. [53] and Gilardi et al. [9] for race 1 and race 4, respectively. Amplicons were analyzed through 1.0% agarose gel electrophoresis, stained with Midori Green Advance gel stain (Nippon, Düren, Germany), and visualized under UV light.

#### 2.2.3. Sequencing and Phylogenetic Analysis

All the sequence data for the *TEF1-α* gene and rDNA-IGS generated in this study were initially visualized using ChromasLite (Technelysium, South Brisbane, Australia) and then compared with a BLAST search on the National Center for Biotechnology Information (NCBI) database to determine the forma specialis of *F. oxysporum*, searching for similarities between the sequences obtained in this study and already existing sequences in the database.

Already published sequences (Table 1) of the four races of *F. oxysporum* f. sp. *lactucae* and other formae speciales of *F. oxysporum* were downloaded and used in the construction of the phylogenetic tree. Phylogenetic analysis was conducted using MEGA version 7.0, and the maximum likelihood (ML) method was used to generate the phylogenetic tree from the concatenated *TEF1-α* gene and rDNA-IGS region. Bootstrap values were obtained from 1000 replicates and only values greater than 60% are shown in the phylogenetic tree. Distances were calculated using Kimura-2p in both phylogenetic inferences. The sequence of *F. subglutinans* was used as an outgroup for rooting the phylogenetic tree.

### 2.3. Pathogenicity Tests

#### 2.3.1. Plant Material

Four representative isolates (GTFus1-4) obtained in this study and other isolates kindly provided from research institutions of Europe and Japan were evaluated for pathogenicity on 12 commercial lettuce cultivars (cvs.) available in the Greek market and on 5 differential lettuce cultivars. Lettuce cultivars, mainly of the Iceberg (syn. Crisphead), Butterhead, Batavia, Romaine (syn. Cos), and Loose-leaf (Oakleaf and Lollo rossa) type, were provided by the breeding company Rijk Zwaan (De Lier, The Netherlands). The differential lettuce cultivars Patriot (susceptible to races 1, 2, and 3 [7]), Costa Rica No. 4 (resistant to race 1 [7], susceptible to race 4 [9]), Banchu Red Fire (resistant to races 2 [7] and 4 [9]), Cavolo di Napoli (susceptible to races 1, 2, and 3 [57] and susceptible to race 4 [9]), and Romana Romabella 30 CN (resistant to races 1 and 2 [57] and susceptible to race 4 [9]) were also included in the pathogenicity tests. Lettuce seedlings were produced in polysterene trays (300 cells/tray) under greenhouse conditions with daily irrigation and without any pesticide application until artificial inoculation.

#### 2.3.2. Inoculation Procedure

Mycelial plugs of six-day-old cultures of the tested isolates were transferred into potato dextrose broth (PDB; Sigma–Aldrich Chemie GmbH, Taufkirchen, Germany) liquid medium and incubated for 10 days at 23 °C with a 12 h photoperiod in a rotary shaker at 100 rpm. Each isolate suspension was passed through a sterilized double-layer cheesecloth, and the inoculum concentration was determined using a hemocytometer to adjust to a density of 1 × 10^6^ cfu/mL.

The inoculation procedure was carried out by removing ten four-week-old seedlings per tested cultivar from the substrate and dipping their roots in the above-described suspension for 5 min. The roots of control plants were dipped solely in sterilized distilled water. The seedlings were then transplanted individually into plastic boxes, each filled with 80 mL of a 5:1 sterilized mixture of peat and perlite. Inoculated and non-inoculated plants were maintained in a growth chamber at 23 °C with a 12/8 h photoperiod cycle and 60% RH for 30 days. Pots were arranged in a randomized complete block design and the experiment was carried out three times.

#### 2.3.3. Estimation of Disease Index

The disease rating was conducted at the end of incubation period. The disease severity index was rated on a scale of 0–4, as described by Gilardi et al. [9]. All data were expressed as disease index (DI) with a scale of 0–100, using the formula:*DI* = [∑(*i* × *ni*)]/(4 × *total of plants*) × 100(1)
with *i* = 0–4 and *n*_i_ = the number of plants with rating *i* [9]. Each value presented in Appendix A is the mean disease index of three replicates ± standard deviation (SD). In addition, according to the 0–100 DI scale, the tested cultivars were characterized in terms of resistance/susceptibility as follows: 0–10, resistant (R); 11–30, partially resistant (PR); 31–60, susceptible (S); and 61–100, highly susceptible (HS).

### 2.4. Defense Gene Expression Measurements

#### 2.4.1. Plant Material for Gene Expression Measurements and RNA Preparation

The pathogenicity assays on the 12 commercial lettuce cultivars revealed that cvs. “Cencibel” and “Lugano” were the most resistant, while cvs. “Sandalina” and “Starfighter” were the most susceptible ones. Thus, these four cultivars were selected for defense-related gene expression measurements, after artificial inoculation with one Greek isolate of the pathogen. The experiment was carried out on four-week-old lettuce plants inoculated with the pathogen, following the same procedure as previously described in Section 2.3.2. For each treatment (artificially inoculated plants and mock-inoculated plants) and respective time point, 5 plants were used, and the whole experiment was repeated three times. Collection of leaf tissue for RNA extraction was carried out at 0, 24, 48, 72, 96, and 168 h post inoculation (hpi). For RNA analysis, each biological sample comprised leaves collected from 15 plants (RNA pooled), and three technical replicates were used for each sample.

Immediately after its removal from the plants, the collected leaf material was immersed in liquid nitrogen and stored afterwards at −80 °C until it was used for further analysis. Total RNA was extracted from 250 mg of tissue using a Nucleo Spin RNA Plant kit (Macherey-Nagel, GmbH & Co. KG, Düren, Germany) according to the manufacturer’s instructions. The concentration of the extracted RNA was measured using a P330 nanophotometer (Implen GmbH, Munich, Germany).

#### 2.4.2. Selection of Defense-Related Genes for Expression Measurements

In total, 10 defense-related genes (*HPL1*, *LTC1*, *SOD*, *PRB1*, *PAL1*, *LOX*, *ERF1*, *MPK*, *BG*, *GST*) were selected for expression measurements in the leaves of the four lettuce cultivars (Appendix A). The selected genes have been associated with defense responses in lettuce after artificial inoculation of the specific host with biotrophic (*Bremia lactucae*) and necrotrophic (*Botrytis cinerea*, *Sclerotinia sclerotiorum*) pathogens in previous studies [35,36].

#### 2.4.3. Quantification of Gene Expression Levels with RT-qPCR

Total RNA, extracted as described above, was used as the template for RT-qPCR. The genes selected for expression measurements and the primers used are listed in Appendix A. The RT-qPCR reactions were carried out in a StepOne Plus Real-Time PCR System (Applied Biosystems, Waltham, MA, USA) using a SYBR Green-based kit (Luna Universal One-Step RT-qPCR Kit, New English Biolabs, Ipswich, MA, USA) according to the manufacturer’s instructions. The amplification conditions were 55 °C for 10 min and 95 °C for 2 min, followed by 40 cycles of 95 °C for 5s and 60 °C for 30 s, while the melt curve stage consisted of 95 °C for 15s, 60 °C for 1 min, and 95 °C for 15s. The threshold cycle (Ct) was determined using the default threshold settings. The 2^−ΔΔCt^ method was applied to calculate the relative gene expression levels [58]. The *Lactuca sativa* actin gene (*Actin*; [36]) was used as the endogenous control, and gene expression levels were expressed in a relative manner with respect to those in non-inoculated lettuce plants at 0 hpi. Analysis of variance was performed with SPSS v25.0 (SPSS Inc., Chicago, IL, USA), and significant differences were determined using Tukey’s multiple range test at the *p* < 0.05 level.

## 3. Results

### 3.1. Pathogen Identification and Phylogenetic Analysis of the Isolates

The *TEF1-α* locus and the IGS region sequences derived from the 84 *F. oxysporum* isolates obtained in this study were queried in the GenBank nucleotide database at the NCBI and compared with other deposited sequences, allowing the identification of the Greek isolates as belonging to *F. oxysporum* f. sp. *lactucae*. The amplification of the *TEF1-α* gene and the IGS region yielded a sequence of approximately 690 and 2300 bp, respectively. The obtained sequences from the BLAST search were identical, so four representative sequences have been deposited in the NCBI database under the accession nos. OQ466113-OQ466116271176 and OQ507251-OQ507254 for *TEF1-α* and IGS, respectively (Table 1). Sequence data of the PCR products were used for the race determination of the Greek isolates, comparing them with *F. oxysporum* f. sp. *lactucae* sequences of all four races and other formae speciales of *F. oxysporum* obtained from the NCBI and published in previous studies (Table 1).

ML analysis was performed based on the combined alignments of *TEF1-α* and IGS, while the sequence data of an *F. subglutinans* isolate (GenBank accession nos. HQ165847.1 (*TEF1-a*); HQ165883.1(IGS)) were used for rooting the phylogenetic tree (Figure 1). The phylogenetic tree was mainly separated into two clades; the first one was composed of the *F. oxysporum* f. sp. *lactucae* race 1 reference isolates, the Greek isolates, and the Dutch isolates (*L. sativa* 04750888, *L. sativa* 04750896) assigned to race 4, supported with a bootstrap of 100%, and the second one was composed only of the race 2 isolates (FK09701, F9501) from Japan. The sequences of isolates MAFF744085 and MAFF744086 (race 3) and those of the remaining formae speciales of *F. oxysporum* retrieved from the NCBI seemed to form individual lineages.

### 3.2. Race Determination with Race 1- and Race 4-Specific PCR Primers

The amplification with primers Hani3′ and Hanilatt3rev, specific to race 1, produced an amplicon of approximately 200 bp in all the Greek isolates and in the previously identified race 1 isolates from Belgium, Norway, and Japan that were used as controls. No amplification product was detected for the Belgian isolate Fus 1.01, the Dutch one, or the Japanese isolates belonging to race 2 and race 3. Using the same isolate collection, the primer pair FUPF and FUPR, specific to the detection of race 4, produced an amplicon of 250 bp only for the Belgian isolate Fus 1.01 and for the Dutch one, which were identified previously as belonging to race 4 [8,9]. However, no amplification was obtained for the Greek isolates used in this study when the same race 4-specific primer pair was used in the PCR assays.

### 3.3. Pathogenicity Test and Race Determination Based on a Set of Differential Lettuce Cultivars

The four representative Greek isolates all induced similar Fusarium wilt symptoms on each of the commercial lettuce cultivars used in this study, indicating that each cultivar reacted as either susceptible or resistant. To give an example, characteristic Fusarium wilt symptoms observed in the pathogenicity assays and the corresponding disease index values used for the evaluation of the disease severity are displayed for two lettuce cultivars in Figure 2.

In total, eight cultivars were characterized as susceptible or highly susceptible, while four were designated as partially resistant or resistant. Based on the disease index (DI) results (Appendix A), “Cencibel” and “Lugano” were the most susceptible cultivars, with DI ranging from 68.6 to 100, while “Sandalina” and Starfighter” were the most resistant ones, with DI ranging from 0.0 to 9.2. The Iceberg and the Lollo Rossa type lettuce cultivars exhibited the same reaction responses (susceptible or highly susceptible) to the artificial inoculation with the Greek *F. oxysporum* f. sp. *lactucae* isolates. The remaining eight cultivars showed a different pathogenic pattern with various types of reaction, rated from susceptible (moderate or high susceptibility) to resistant (partial or high resistance) when evaluating them in pairs of the same lettuce type (Butterhead, Batavia, Oakleaf, or Romaine), as is depicted in the heatmap presented in Figure 3. The two cultivars used for each of the Butterhead, Batavia, Oakleaf, and Romaine types showed inconsistent resistance levels. For instance, cv. “Tacitus” (Romaine type) was partially resistant or resistant, while cv. “Tanius” of the same type was more susceptible to the artificial inoculation with the pathogen.

The reaction of the differential lettuce cultivars “Cavolo di Napoli” and “Patriot” was similar, as they were both susceptible to the four Greek isolates, exhibiting DI ranging from 51.9 to 66.8, and from 50.0 to 62.0, respectively. In addition, “Costa Rica No. 4” was found to be resistant (DI from 0.0 to 5.6) and “Banchu Red Fire” susceptible (DI from 32.8 to 53.8). On the other hand, low DI values (from 0.0 to 12.9) were recorded when the cultivar “Romana Romabella 30 CN” was evaluated for pathogenic reaction to the Greek isolates. Similar virulence was observed when the race 1 reference isolates from Japan and Belgium (MAFF 244120 and Fus 1.39, respectively) were tested for pathogenicity on the five differential lettuce cultivars. In contrast, the reference race 1 isolates showed divergent pathogenicity reactions in some cases, compared with the one observed when the Greek isolates were tested. For instance, cv. “Cencibel” appeared to be highly susceptible when the Greek isolates were tested and susceptible or unexpectedly resistant, in the case of isolate Fus 1.39 and MAFF 244120, respectively (Figure 2).

The reference race 2 and race 3 isolates showed comparable virulence on the differential set of cultivars to that described in previous studies, while the race 4 isolates showed almost the same virulence. Cultivars “Cavolo di Napoli”, “Patriot”, “Costa Rica No. 4”, and “Romana Romabella 30 CN” were susceptible or highly susceptible, while “Banchu Red Fire” was partially resistant to the two isolates (04750888 and Fus 1.01) assigned to the newly described race 4 (Appendix A).

### 3.4. Defense Genes’ Expression in the Resistant and Susceptible Lettuce Cultivars

To determine whether the observed differences in cultivar susceptibility were related to biochemical resistance mechanisms, the expression of 10 defense-related genes (*HPL1*, *LTC1*, *SOD*, *PRB1*, *PAL1*, *LOX*, *ERF1*, *MPK*, *BG*, *GST*) was measured in artificially inoculated lettuce plants of the two most susceptible (cvs. “Cencibel” and “Lugano”) and the two most resistant (cvs. “Sandalina” and “Starfighter”) cultivars. Data on gene expression measurements are shown in Figure 4.

The expression profiles of selected genes were surveyed at five different time points (24, 48, 72, 96, and 168 hpi). *HPL1* was up-regulated in the resistant cultivars compared to the expression levels observed in the susceptible ones, reaching a peak at 48 hpi, and showing a 6.25-fold increase at that time point comparing the highest and lowest expression values. However, although *HPL1* was still up-regulated, its transcription started to decline thereafter. *LTC1* was highly expressed at 72 and 96 hpi for the two resistant cultivars, showing an 82.3- and 51.5-fold increase, while the highest induction levels for *PRB1* and *BG* were observed at 24 hpi (3.0- and 3.8-fold changes, respectively), and then their expression levels decreased for all the evaluated cultivars. The highest induction level for the superoxide dismutase (*SOD*) gene was observed at 24 hpi, after which it started to decrease. However, in the later stages of infection, it presented significantly higher induction levels for the resistant cultivars in comparison to the susceptible ones. Phenylalanine ammonia lyase (*PAL1*) and lipoxygenase (*LOX*) exhibited a similar bimodal expression pattern, reaching a peak at 24 and 48 hpi in the resistant cultivars compared to that of the susceptible ones, with 7.8- and 12.7- (*PAL*), and 7.1- and 12.8-fold changes (*LOX*), respectively. On the other hand, the highest transcript levels for *ERF1*, with a 10.5-fold change, were observed 24 hpi, while at the later time points, this gene showed decreasing expression levels. Interestingly, *MPK* and *GST* were significantly up-regulated in the resistant cultivars compared to the susceptible ones across the different time points, exhibiting their highest induction 72 and 48 hpi, with 4.1- and 2.6-fold increases, respectively.

## 4. Discussion and Conclusions

The present paper represents the first comprehensive study that has been conducted aiming to identify the pathogen associated with Fusarium wilt symptoms in lettuce in Greece. The results of this study revealed that the Fusarium wilt disease of lettuce is caused by *Fusarium oxysporum* f. sp. *lactucae*, assigning the obtained isolates to race 1. In Europe, race 1 is the most widespread, with reports from several countries [3,4,59,60].

Our results from the pathogenicity tests showed that cultivars “Cavolo di Napoli”, “Costa Rica No. 4”, and “Banchu Red Fire” were susceptible to highly susceptible to the Greek *F. oxysporum* f. sp. *lactucae* isolates, while cvs. “Costa Rica No. 4” and “Romana Romabella 30 CN” were resistant to them. These results, as well as the reaction results of a set of isolates belonging to the four identified races of the pathogen, were comparable to those derived from the pathogenicity assays carried out in previous studies [6,7,8,9,12,53], indicating that the isolates obtained in this study could be assigned to race 1.

The pathogenic characteristics of 12 of the most-cultivated lettuce cultivars in Greece have also been evaluated. Specifically, these cultivars exhibited variable levels of resistance or susceptibility to the pathogen, in some cases even for cultivars of the same type. For instance, data obtained in this assay showed that cultivars from the Iceberg or Lollo Rossa type were highly susceptible to the pathogen, indicating that such lettuce cultivars should not be selected for cultivation in fields where the disease is present. Furthermore, the differences observed in the disease severity index values among the same combination (Greek isolates × lettuce cultivars) could be attributed to the different virulence of the isolates or to the genetic characteristics of the cultivars used, according to Scott et al. [61] and Gordon and Koike [62]. Such inconsistent associations between lettuce type and reaction to the pathogen have been described before for cultivars of the Romaine [61] and Butterhead type [53].

Our results from sequence analysis showed that the sequences of the translation elongation factor 1-alpha (*TEF1-α*) gene and the rDNA intergenic spacer (IGS) region were identical for all the 84 Greek isolates of *F. oxysporum* f. sp. *lactucae* examined in this study. Sequences obtained for these two loci have been also found to be identical in previous studies, suggesting that limited genetic diversity is detected among isolates associated with a given race of *F. oxysporum* f. sp. *lactucae* [9,14,16,20,54]. In any case, sequences of the *TEF1-a* gene and the rDNA-IGS region are commonly used for the comparative analysis of isolates belonging to the genus *Fusarium* [9,16,55,63], and applied in phylogenetic analyses [55].

The phylogenetic analysis using the concatenated sequences of the *TEF1-a* gene and the IGS region grouped the Greek isolates into the same clade, with other isolates assigned to race 1 and with those assigned recently to race 4. The sequences of isolates assigned to races 1 and 4 were also grouped in one sole clade in the study by Gilardi et al. [9], who identified race 4 as a new race of *F. oxysporum* f. sp. *lactucae,* and later in the study by Cabral et al. [54]. Moreover, in our study, sequences of race 2 formed a separate clade, confirming the previous results that noted that races 1 and 2 are separated into two different genetic lineages [16]. Furthermore, isolates belonging to races 1, 2, and 3 are genetically dissimilar, grouped in different clades, thus implying that they most likely have different origins [14]. Isolates corresponding to races 1, 2, and 3 might have evolved independently from different ancestors and exhibit variable resistance reactions on differential lettuce cultivars, a differential system which is widely used to identify pathogenic races of the specific pathogen [7]. Isolates belonging to race 3 and to other formae speciales formed individual lineages, demonstrating the low resolution ability of the *TEF1-a* gene and rDNA-IGS region to distinguish different formae speciales of *F. oxysporum* [55]. Thus, besides pathogenicity tests, additional molecular tools are needed for the discrimination of isolates assigned to race 1 and race 4, which are the two races identified so far in Europe. Hence, PCR assays with specific primers for race 1 [5] and race 4 [9] were also performed. PCR provided amplicons of the expected size when race 1-specific primers were used for the amplification of DNA of the Greek isolates, while no positive results were detected in the case of race 4-specific primers.

To shed light on the biochemical basis of the observed resistance of cvs. “Sandalina” and “Starfighter” or the susceptibility of cvs. “Cencibel” and “Lugano” to Fusarium wilt, plants of these four cultivars were artificially inoculated with the pathogen, and the expression of 10 defense-related genes (*HPL1*, *LTC1*, *SOD*, *PRB1*, *PAL1*, *LOX*, *ERF1*, *MPK*, *BG*, *GST*) was analyzed in leaf tissue collected at five different time points post inoculation (24, 48, 72, 96, and 168 hpi). Salicylic acid (SA), jasmonic acid (JA), and ethylene (ET) are the plant hormones that play a key role to the regulation of plant defense responses, as they are produced in higher levels in plant tissues after pathogen infection. Although SA signaling provides plant resistance to biotrophic and hemibiotrophic pathogens, while the JA/ET signaling pathways enhance resistance to necrotrophic ones, this is something of a generalization [64].

Temporal and quantitative gene expression fluctuations were observed between the resistant and the susceptible cultivars as a reaction to the pathogen infection. Some genes, such as *PRB1*, *SOD*, *ERF1*, *PAL1*, *LOX*, and *BG,* were up-regulated at the earliest stage of infection (24 hpi) in the resistant cultivars in comparison with the susceptible ones, while the remaining genes showed their highest expression levels 48 (*HPL1*, *GST*) or 72 (*LTC1*, *MPK*) hpi. However, some genes (*SOD*, *ERF1*, *MPK*, and *GST*) showed significant expression levels across the five evaluated time points. Almatwari et al. [36] are of the opinion that such differentiations in the expression levels of defense-related genes might be attributed to *L. sativa* interactions with pathogens, preventing disease development. More specifically, the induction of the defense-related genes *PRB1*, *SOD*, *ERF1*, *LTC1*, and *HPL1* was found to reach the highest levels of induction 24 hpi in the resistant lettuce cultivar, demonstrating the importance of their expression to the mediated resistance towards *S. sclerotiorum* infection [36]. A previous study, aiming to assess the response of *PRB1*, *PAL*, *LOX*, *ERF1*, *MPK*, *HPL1*, *BG*, and GST genes, among other genes, to the necrotrophic pathogen *Botrytis cinerea* and to the biotrophic pathogen *Bremia lactucae*, revealed that these genes are significantly induced for both pathogens, implying the induction of similar pathways regardless of the pathogen cycle [35].

In our study, the responses of lettuce plants at the earliest stage after artificial inoculation with the pathogen (24 hpi) revealed an overexpression of genes such as *ERF1*, *PAL1*, and *LOX*, which are associated with JA/ET signaling. In addition, the *HPL1* and *LTC1* genes, of the same signaling pathway, showed their highest expression levels 48 and 72 hpi, respectively. On the other hand, marker genes of the SA signaling pathway such as *PRB1* and *BG* showed their highest induction levels 24 hpi. Interestingly, the *SOD*, *ERF1*, *MPK*, and *GST* genes were induced at all the evaluated time points. However, these genes exhibited their highest induction at different timepoints; *SOD* and *ERF1* 24 hpi, *GST* 48 hpi, and *MPK* 72 hpi. These findings show a synergistic activation of both the SA and JA/ET signaling pathways upon inoculation with the pathogen *F. oxysporum* f. sp. *lactucae*, particularly in the resistant cultivars, confirming that resistance to *F. oxysporum* is a genetically complex trait [30]. A similar crosstalk between these two signaling pathways has also been reported in previous studies on the interactions observed between lettuce plants and various pathogens, such as *B. cinerea*, *B. lactucae,* and *S. sclerotiorum* [35,36].

In conclusion, wilt symptoms in lettuce plants in Greece have been associated with race 1 *F. oxysporum* f. sp. *lactucae* based on a combination of molecular identification and pathogenicity on a set of differential lettuce cultivars. It is also deduced that some of the commonly used lettuce cultivars in this country exhibit different levels of resistance to race 1 of this pathogen. This observation is important as the use of resistant cultivars is considered as the main method of controlling Fusarium wilt of lettuce [62,65]. However, an integrated management program should be applied in commercial cultivations as it is rather difficult to control Fusarium wilt solely with resistant cultivars [3,62,66]. Such a program could include healthy seed and propagation material, as well as crop rotation [3]. However, the latter management approach is questionable since the pathogen can survive in the soil for long periods, even after rotation with plant species that are not hosts of the specific forma specialis [20,67,68]. Race 1 isolates of the pathogen tend to cause more severe disease symptoms under warmer conditions [61,68], an epidemiological aspect that could be of great importance in the context of climate change. On the other hand, Fusarium wilt symptoms that have already been assigned to race 4 in several European countries are favored in lower temperatures, thus explaining the geographical distribution of this race of the pathogen so far [69].

## Figures and Tables

**Figure 1 microorganisms-11-01082-f001:**
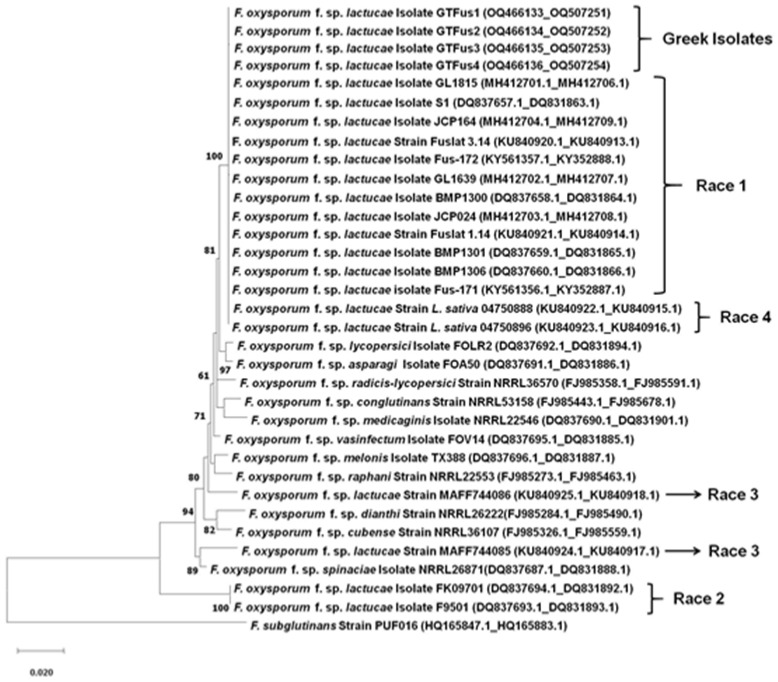
Phylogenetic tree of representative Greek *F. oxysporum* f. sp. *lactucae* isolates, obtained from lettuce plants exhibiting Fusarium wilt symptoms, constructed using maximum likelihood (ML) analysis, based on the concatenated sequences of their translation elongation factor 1-alpha (*TEF1-α*) gene and the ribosomal DNA intergenic spacer (rDNA-IGS) region. Concatenated sequences from *F. subglutinans* were used as the outgroup to root the tree. Genetic distances were determined according to Kimura’s substitution model, and bootstrap support was estimated based on 1000 trials. Only values over 60% are shown at the nodes. Branch length is proportional to the number of nucleotide substitutions, as indicated by the scale bar. Scale bar represents one base change per 50 nucleotide positions. Numbers in parenthesis are the accession nos. of sequences obtained from the National Center for Biotechnology Information (NCBI).

**Figure 2 microorganisms-11-01082-f002:**
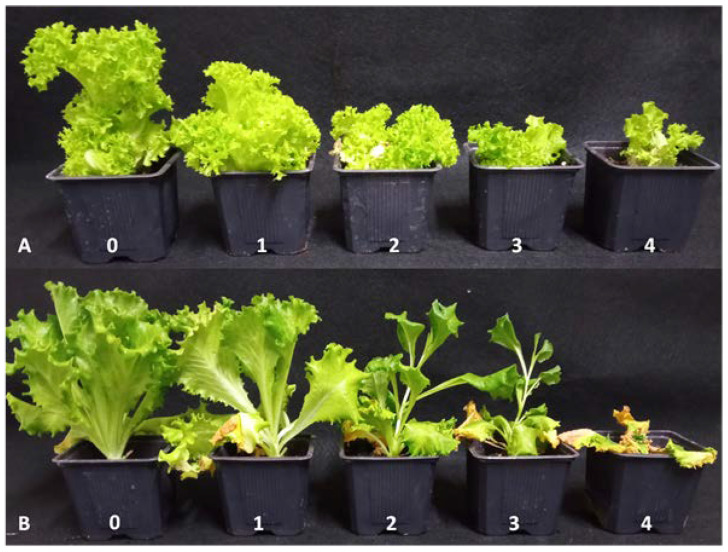
Disease index scale (0–4) used for evaluation of Fusarium wilt severity on (**A**) “Lugano” and (**B**) “Sandalina” lettuce cultivars artificially inoculated with Greek *F. oxysporum* f. sp. *lactucae* isolates, where 0 = healthy plant; 1 = initial symptoms of leaf chlorosis, slight reduction in development, slight vascular browning; 2 = severe leaf chlorosis, evident reduction in development, sometimes asymmetric development of the head, evident vascular browning; 3 = leaf chlorosis and inhibition of growth, evident deformation and initial vascular browning symptoms of wilting during the hottest hours of the day; and 4 = plant strongly deformed with leaf chlorosis or completely necrotic leaves, totally wilted. Disease index scale symptoms depicted in the figure correspond to characteristic symptoms recorded throughout the pathogenicity testing with the different isolates for the same cultivar.

**Figure 3 microorganisms-11-01082-f003:**
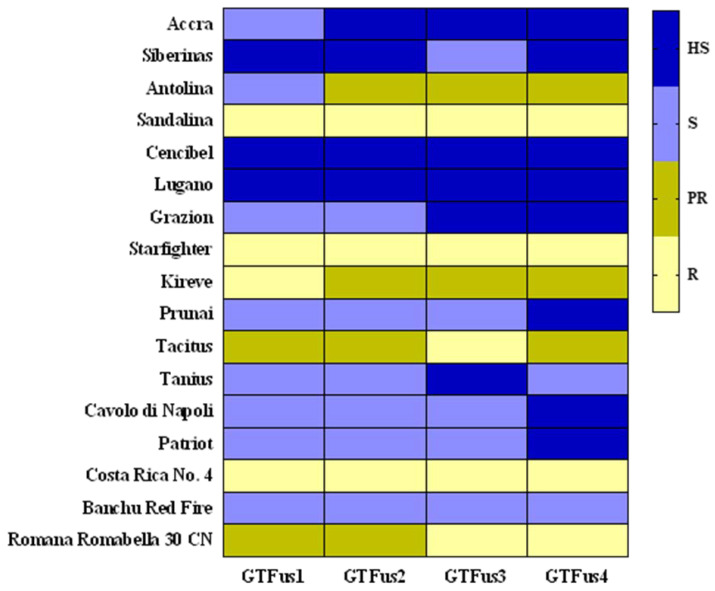
Reaction of commercial and differential lettuce cultivars to the artificial inoculation with the four Greek *F. oxysporum* f. sp. *lactucae* isolates (GTFus1-4) used in this study. The heatmap was constructed based on the pathogenicity test results expressed as a disease index of 0 to 100 and depicted in terms of resistance–susceptibility. Resistant cultivars (R) are indicated by a pale yellow color, partially resistant (PR) by olive green, susceptible (S) by light blue, and the highly susceptible ones (HS) are indicated by a dark blue color, as explained by the color scale at the right side of the figure.

**Figure 4 microorganisms-11-01082-f004:**
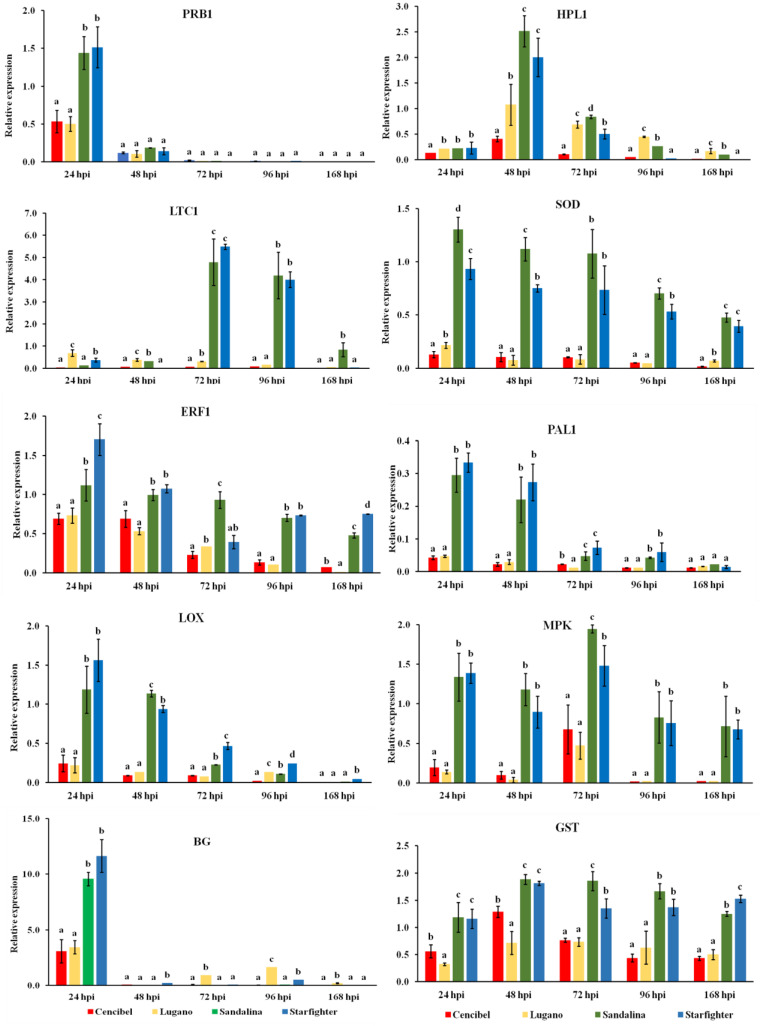
Expression analysis of defense-associated genes in lettuce plant leaves of two susceptible (cvs. “Cencibel” and “Lugano”) and two resistant (cvs. “Sandalina” and “Starfighter”) cultivars through real-time quantitative PCR (RT-qPCR) at 12, 24, 48, 96, and 168 h post inoculation (hpi) with *F. oxysporum* f. sp. *lactucae*. The *y*-axis represents relative differences in gene expression compared to that of lettuce plants before *F. oxysporum* f. sp. *lactucae* inoculation (time point 0 h). cDNA samples were normalized using the *Actin* gene as endogenous control. Each column represents average data with error bars from three independent technical samples. Different letters in the columns indicate significant differences between the four cultivars for the same gene at each time point according to Tukey’s test (*p* < 0.05).

**Table 1 microorganisms-11-01082-t001:** Isolates and their corresponding accession numbers used for the phylogenetic analysis in this study.

Isolate Name	Species Name	Geographical Origin	GenBank Accession Numbers ^x^		Ref.
			*TEF1-α*	IGS	
GTFus1	*Fusarium oxysporum* f. sp. *lactucae* race 1	Greece	OQ466113	OQ507251	*
GTFus2	*F. oxysporum* f. sp. *lactucae* race 1	Greece	OQ466114	OQ507252	
GTFus3	*F. oxysporum* f. sp. *lactucae* race 1	Greece	OQ466115	OQ507253	
GTFus4	*F. oxysporum* f. sp. *lactucae* race 1	Greece	OQ466116	OQ507254	
Fuslat 1.14	*F. oxysporum* f. sp. *lactucae* race 1	Italy	KU840921.1	KU840914.1	[9]
Fuslat 3.14	*F. oxysporum* f. sp. *lactucae* race 1	Italy	KU840820.1	KU840913.1	
GL1815	*F. oxysporum* f. sp. *lactucae* race 1	California, USA	MH412701.1	MH412706.1	[20]
GL1639	*F. oxysporum* f. sp. *lactucae* race 1	California, USA	MH412702.1	MH412707.1	
JCP024	*F. oxysporum* f. sp. *lactucae* race 1	California, USA	MH412703.1	MH412708.1	
JCP164	*F. oxysporum* f. sp. *lactucae* race 1	Arizona, USA	MH412704.1	MH412709.1	
BMP1300	*F. oxysporum* f. sp. *lactucae* race 1	Arizona, USA	DQ837658.1	DQ831864.1	[16]
BMP1301	*F. oxysporum* f. sp. *lactucae* race 1	Arizona, USA	DQ837659.1	DQ831865.1	
BMP1306	*F. oxysporum* f. sp. *lactucae* race 1	Arizona, USA	DQ837660.1	DQ831866.1	
Fus-171	*F. oxysporum* f. sp. *lactucae* race 1	Brazil	KY561356.1	KY352887.1	[54]
Fus-172	*F. oxysporum* f. sp. *lactucae* race 1	Brazil	KY561357.1	KY352888.1	
S1	*F. oxysporum* f. sp. *lactucae* race 1	Japan	DQ837657.1	DQ831863.1	[16]
F9501	*F. oxysporum* f. sp. *lactucae* race 2	Japan	DQ837693.1	DQ831893.1	
FK09701	*F. oxysporum* f. sp. *lactucae* race 2	Japan	DQ837694.1	DQ831892.1	
MAFF744085	*F. oxysporum* f. sp. *lactucae* race 3	Japan	KU840924.1	KU840917.1	[9]
MAFF744086	*F. oxysporum* f. sp. *lactucae* race 3	Japan	KU840925.1	KU840918.1	
*L. sativa* 04750888	*F. oxysporum* f. sp. *lactucae* race 4	Netherlands	KU840922.1	KU840915.1	
*L. sativa* 04750896	*F. oxysporum* f. sp. *lactucae* race 4	Netherlands	KU840923.1	KU840916.1	
FOV14	*F. oxysporum* f. sp. *vasinfectum*	California, USA	DQ837695.1	DQ831885.1	[16]
FOLR2	*F. oxysporum* f. sp. *lycopersici*	California, USA	DQ837692.1	DQ831894.1	
TX388	*F. oxysporum* f. sp. *melonis*	Texas, USA	DQ837696.1	DQ83188.1	
FOA50	*F. oxysporum* f. sp. *asparagi*	Australia	DQ837691.1	DQ831886.1	
NRRL22546	*F. oxysporum* f. sp. *medicaginis*	SE Asia	DQ837690.1	DQ831901.1	
NRRL26871	*F. oxysporum* f. sp. *spinaciae*	Japan	DQ837687.1	DQ831888.1	
NRRL53158	*F. oxysporum* f. sp. *conglutinans*	North Carolina, USA	FJ985443.1	FJ985678.1	[55]
NRRL26222	*F. oxysporum* f. sp. *dianthi*	Israel	FJ985284.1	FJ985490.1	
NRRL36570	*F. oxysporum* f. sp. *radicis*-*lycopersici*	Unknown	FJ985358.1	FJ985591.1	
NRRL36107	*F. oxysporum* f. sp. *cubense*	Honduras	FJ985326.1	FJ985559.1	
NRRL22553	*F. oxysporum* f. sp. *raphani*	Germany	FJ985273.1	FJ985463.1	
PUF016	*F. subglutinans*	China	HQ165847.1	HQ165883.1	[56]

^x^ Accession numbers of sequences of representative isolates belonging to several *Fusarium oxysporum* formae speciales obtained from the National Center for Biotechnology Information (NCBI). * Representative isolates obtained in this study and used for the phylogenetic analysis.

## Data Availability

The data presented in this study are available in this article.

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
