# Peer review of "Identification of Fusarium oxysporum f. sp. lactucae Race 1 as the Causal Agent of Lettuce Fusarium Wilt in Greece, Commercial Cultivars’ Susceptibility, and Temporal Expression of Defense-Related Genes"

_microorganisms, 2023, doi:10.3390/microorganisms11041082_

Round 1
Reviewer 1 Report
Please accept the paper once address the following queries.
1. Why do race 2 and race 3 come with other formae sp of F. oxysporum in clade 2? Whether the host range of race 2 and race 3 is studied to decipher their real position within F.oxysporum form a specialist group, rather than telling polyphyletic nature of F. oxysporum f.sp. lactucae
2. Disease reaction in Fig 3 should be given as per the rating scale; the heatmap creates more confusion in this case. Kindly change it 3. Are the symptoms produced in the 'Lugano' and 'Sandalina' cultivars caused by the same isolate of Fusarium oxy. f.sp. lactucae? As totally distinct symptoms were observed in two different sets in Figure 2. 4. Regarding the expression level of ERF 1, statements in lines 402 and 496 contradict each otherAuthor Response
Responses to Reviewer 1
The authors would like to thank the reviewer for the valuable suggestions and ideas that increased the quality of the manuscript. We really appreciate the fact the reviewer gave us the opportunity to present our work in the ''Microorganisms'' Journal.
Below you can find a list of our response to each point raised. We have incorporated these comments into the manuscript, and we thus hope that the manuscript can be accepted for final publication in ''Microorganisms''.
All the suggested changes have been highlighted within the document by using the TRACK CHANGES MODE.
Comment 1: Why do race 2 and race 3 come with other formae sp of F. oxysporum in clade 2? Whether the host range of race 2 and race 3 is studied to decipher their real position within F.oxysporum form a specialist group, rather than telling polyphyletic nature of F. oxysporum f.sp. lactucae
Response: The results indicate that the 3 races are genetically quite different. More specifically, isolates belonging to race 2 and 3 are distributed at random among other formae spaciales of F. oxysporum. An explanation has been added in the Discussion Section of the manuscript.
Comment 2: Disease reaction in Fig 3 should be given as per the rating scale; the heatmap creates more confusion in this case. Kindly change it
Response: Figure 3 has been changed to present the results of the pathogenicity test with the Greek isolates based on the disease severity index values and expressed in terms of resistance – susceptibility.
Comment 3: Are the symptoms produced in the 'Lugano' and 'Sandalina' cultivars caused by the same isolate of Fusarium oxy. f.sp. lactucae? As totally distinct symptoms were observed in two different sets in Figure 2.
Response: Disease index scale symptoms pictured in the Figure correspond to characteristics symptoms recorded throughout the pathogenicity test with the different isolates for the same cultivar.
Comment 4: Regarding the expression level of ERF 1, statements in lines 402 and 496 contradict each other
Response: An additional sentence has been added to be more precise and informatory.
Reviewer 2 Report
My revision report is attached herewith including the revise file.

Author Response
Responses to Reviewer 2
The authors would like to thank the reviewer for the valuable suggestions and ideas that increased the quality of the manuscript. We really appreciate the fact the reviewer gave us the opportunity to present our work in the ''Microorganisms'' Journal.
Below you can find a list of our response to each point raised. We have incorporated these comments into the manuscript, and we thus hope that the manuscript can be accepted for final publication in ''Microorganisms''.
All the suggested changes have been highlighted within the document by using the TRACK CHANGES MODE.
Comment 1: ABSTRACT
It lacks both introductive and conclusive remarked sentences that could render the paper more attractive for reader. Other little adjustments are highlighted in the revise file.
Response: Introductive and conclusive sentences have been added in the Abstract. In addition, we have addressed and incorporated the highlighted adjustments in the revised manuscript.
Comment 2: INTRODUCTION
L.64-79. In my opinion, this section should instead introduce the paper. Thus, I think that it should be moved at the beginning of the paragraph. Other little adjustments are highlighted in the revise file.
Response: We would rather keep the specific paragraph at this part of the Introduction as it is consistent with the way we wish to present our work and thus this paragraph precedes the gene expression part of the Introduction section. In addition, we have addressed and incorporated the highlighted adjustments in the revised manuscript.
Comment 3: MATERIALS AND METHODS
Little adjustments are highlighted in the revise file.
Response: We have addressed and incorporated the highlighted adjustments in the revised manuscript.
Comment 4: RESULTS
-Figure 2 should be moved in supplementary material and re-named as Figure S1.
-It would be useful to draw up a new graph (figure) as replacement of figure 2 in which the cultivars are shown on the x-axis and the disease severity index on the y-axis with suitable statistical elaboration (error bars, Tukey’s test, etc.). Other little adjustments are highlighted in the revise file.
Response: We maintained Figure 2 in the manuscript. Furthermore, following the suggestion made by Reviewer 1 we changed Figure 3 to present the results of the pathogenicity test with the Greek isolates based on the disease severity index values and expressed in terms resistance–susceptibility. Thus, the results of the pathogenicity test for the Greek isolates are presented in Figure 3, as well as, in Table S1. In addition, we have addressed and incorporated the highlighted adjustments in the revised manuscript.
Comment 5: DISCUSSION
-L.411-423. This section seems partially overlapped with the Introduction section. Please, check it and solve accordingly.
Response: We have edited down the specific part of the manuscript, deleting the overlapping sentences.
